# Application of C-LSTM Networks to Automatic Labeling of Vehicle Dynamic Response Data for Bridges

**DOI:** 10.3390/s22093486

**Published:** 2022-05-03

**Authors:** Ryota Shin, Yukihiko Okada, Kyosuke Yamamoto

**Affiliations:** 1Graduate School of Systems and Information Engineering, University of Tsukuba, 1-1-1 Tennodai, Tsukuba 305-8577, Ibaraki, Japan; shin.ryota.sp@alumni.tsukuba.ac.jp; 2Faculty of Engineering, Information and Systems, University of Tsukuba, 1-1-1 Tennodai, Tsukuba 305-8577, Ibaraki, Japan; okayu@sk.tsukuba.ac.jp; 3Center for Artificial Intelligence Research, University of Tsukuba, 1-1-1 Tennodai, Tsukuba 305-8577, Ibaraki, Japan

**Keywords:** drive-by bridge monitoring, vehicle bridge interaction, neural network, C-LSTM, field test

## Abstract

Maintaining bridges that support road infrastructure is critical to the economy and human life. Structural health monitoring of bridges using vibration includes direct monitoring and drive-by monitoring. Drive-by monitoring uses a vehicle equipped with accelerometers to drive over bridges and estimates the bridge’s health from the vehicle vibration obtained. In this study, we attempt to identify the driving segments on bridges in the vehicle vibration data for the practical application of drive-by monitoring. We developed an in-vehicle sensor system that can measure three-dimensional behavior, and we propose a new problem of identifying the driving segment of vehicle vibration on a bridge from data measured in a field experiment. The “on a bridge” label was assigned based on the peaks in the vehicle vibration when running at joints. A supervised binary classification model using C-LSTM (Convolution—Long-Term Short Memory) networks was constructed and applied to data measured, and the model was successfully constructed with high accuracy. The challenge is to build a model that can be applied to bridges where joints do not exist. Therefore, future work is needed to propose a running label on bridges based on bridge vibration and extend the model to a multi-class model.

## 1. Introduction

Bridges are critical infrastructure structures that support transportation networks. Bridges daily degrade due to varied external factors, such as deterioration caused by chemical reactions with oxygen and chloride ions in the air, high cycle fatigue caused by traffic vibrations, loss of members due to impact loads of traffic accidents, and large-scale destruction caused by natural disasters. The economic impact of bridge malfunction is significant and sometimes life-threatening. Therefore, early detection of structural performance deterioration of bridges is essential, and appropriate monitoring and maintenance are crucial. One method of vibration monitoring in bridges is installing accelerometers directly on the bridge. While this method can evaluate bridge performance with high accuracy, there are challenges in applying it to multiple bridges. The need to install multiple sensors on a single bridge is time-consuming, costly, problematic in securing a power source, requires traffic control during installation, and is sometimes hazardous.

Yang et al. (2004) [1] proposed mounting sensors on traveling vehicles instead of the target bridge. Since the vehicle vibration contains the bridge’s mechanical information, it is expected to estimate the bridge condition indirectly from the vehicle vibration data. For example, Lin and Yang (2005) [2] experimentally extracted the bridge’s natural frequency from the Fourier’s power spectrum of the measured vehicle vibration. Yang and Chang (2009) [3,4] tried to estimate the bridge’s natural frequencies accurately for considering its application to bridge damage detection.

Inspired by Yang’s indirect approach, many researchers have vigorously developed varied drive-by monitoring technologies. Xiang et al. (2010) [5] applied STFT (Short-Time Fourier Transform) to vibration data of a traveling vehicle with a shaker to evaluate the bridge condition. Their numerical examination shows that the power of vehicle vibration in the time-frequency domain reacts sensitively to the bridge’s local damage. Nguyen et al. (2010) [6] also showed a similar result by applying wavelet transform. According to their results, it is efficient to use the spatial indices of the bridge vibration to detect damage. Thus, Oshima et al. (2014) [7] proposed a method to estimate the bridge’s mode shape from vehicle vibrations. However, these drive-by monitoring technologies based on spatial indices assume that the vehicle position can be measured accurately. Takahashi et al. (2019) [8] used SSMA (Spatial Singular-Mode Angle) for damage detection and also confirmed its high sensitivity, while the SSMA-based method also requires accurate vehicle position data temporally synchronized with the vehicle vibration data.

The data measured in the actual vehicles passing through the actual bridges contain various noises caused by varied factors—for example, the effects of road surface roughness, vehicle speed, and temperature. Malekjafarian et al. (2019, 2020) [9,10] used artificial neural networks and Gaussian processes to analyze the relationship between these factors and vehicle response analysis. Corbally and Malekjafarian (2021) [11] showed that the model above can be used to detect changes in boundary conditions due to cracks in the bridge deck and bearing seizures. Locke et al. (2020) [12] showed that bridge damage discrimination using vehicle vibration can be estimated with high accuracy even when environmental and operational noise, such as road traffic, road surface roughness, and temperature, are included. Sarwar and Cantero (2021) [13] suggested a method to estimate the location and extent of bridge damage using deep autoencoders from vehicle vibration. As environmental factors, the number of vehicles, vehicle speed, measurement noise, and variation in vehicle characteristics are considered, and robust estimation results can be obtained for each of them. Although these studies are based on vehicle vibrations simulated numerically, they suggest the possibility of bridge damage detection by drive-by monitoring.

Although many papers explicitly state that they measure vehicle vibration, many also assume the use of vehicle position in their signal processing [7,8]. Satellite positioning, navigation, and timing systems, the so-called GPS (Global Positioning System), can measure a vehicle’s position. However, the readily available GPS devices are also subject to significant errors, so a vehicle’s travel position is not reliable. Bridge locations can be obtained from GIS (Geographic Information System) data, but the resolution is too coarse. Accurate information on the location of each bridge is usually unknown or not digitized. Furthermore, drive-by monitoring primarily uses data only while a vehicle passes over a bridge. Because the time that a vehicle travels on a bridge is so short, the location information measured by GPS devices is limited to a rough extraction of interest data. Therefore, when driving data on bridges are extracted from GPS-only location data, it is likely to include data from sections of the bridge that are not bridges, and this will likely harm the results of the analysis. In addition, multiple runs are a promising approach for drive-by monitoring to overcome the problems of road surface irregularities, environmental effects, and limited measurement time [14]. Therefore, for the social implementation of drive-by monitoring, it is essential to have a technology that can automatically extract only the target data from considerable driving data.

As a possible idea about signal processing to extract the “on a bridge” vehicle response, the method proposed by Wang et al. (2018) [15] can be considered. This technique is for estimating the natural bridge frequencies from vehicle vibration. A particle filter (sequential Monte Carlo method) is used to estimate the input profile from the vehicle vibration. The input profile is expressed as the sum of the road surface roughness and bridge vibration at the axle position. Assuming that the vehicle is moving in a straight line, the road profiles of the front and rear wheels are the same. The bridge vibration component can be obtained as the difference between the input profiles of the front and rear wheels. However, this method assumes that the vehicle characteristics are given in advance. Murai et al. (2019) [16] succeeded in extracting “on a bridge” signals by using the difference of the unsprung mass vibration of the vehicle. This method is verified by numerical simulation. Shin et al. (2021) [17], who examined this method in a field test, pointed out that it is difficult to visually determine the “on a bridge” section due to the influence of environmental uncertainties and measurement noise. However, Shin et al. also showed that C-LSTM networks, which learn from temporal and spatial features, can estimate entry and exit timing. However, utilizing “on a bridge” labels remains a challenge. Because of the low reliability of the GPS devices, it is not easy to eliminate the influence of the road profile. Therefore, this study investigates automatic labeling of the vehicle response to a bridge. The data are measured on the actual vehicle. The models are constructed separately by using the data obtained at each axle.

Neural networks are used in various fields, most notably in speech recognition and natural language processing, where they have shown excellent performance (Donahue et al., 2015 [18]). CNN (Convolutional Neural Network) has the advantage of learning local responses from spatial-domain–time-domain data, but they are not suitable for learning sequential correlations. On the other hand, RNN (Recurrent Neural Network) can be characterized for sequential modeling but is not suitable for parallel processing (Zhou et al., 2015 [19]). In addition, for the gradient exploding or vanishing problem caused by general RNNs, a LSTM unit can be introduced to analyze the data, including past information (Hochreiter and Schmidhuber [20]). The combination of CNN and LSTM can be used for visual recognition and description (Donahue et al., 2015 [18]), text classification (Zhou et al., 2015 [19]), web traffic anomaly detection (Kim and Cho, 2018 [21]), and residential energy consumption prediction (Kim and Cho, 2019 [22]). Acceleration data measured by vehicles traveling on bridges are time-series data that contain both temporal and spatial features. In recent years, it has been applied to direct bridge damage detection (Yang et al., 2020 [23], Yang et al., 2021 [24]), and we thought that it would be possible to create a model that discriminates only the portion of the vehicle acceleration data traveling on the bridge by using C-LSTM.

Wang et al. (2019) [25] proposed estimating tire ground forces with relatively high accuracy even at vehicle speeds of about 30 km/h using a Kalman filter based on vehicle vibrations measured with an iPod installed on the vehicle. Yang et al. (2020) [26] compared the accuracy of estimating the natural frequencies of bridges using data measured by sensors installed on special vehicles and directly measured bridge vibration. When traveling at relatively low speeds, it is possible to estimate up to the second-order natural frequencies of bridges without being affected by the frequency of the vehicle by determining the ground point force from the vehicle vibration. Yang et al. (2022) [27] proposed a comprehensive indirect method for estimating girder bridges’ fundamental mode dynamic properties with two identical trailer-mounted passing tractors and evaluating the element stiffness. They were validated both numerically and in field experiments. Locke et al. (2022) [28] compared the system identification capabilities of different OMA (Operational Modal Analysis) techniques on bridges less than 18.28 m (60 ft). The experimental results demonstrate that OMA techniques can be utilized to correctly identify the frequencies of short-span bridges in the dynamic response of a passenger vehicle even in the presence of time-varying and nonlinear system characteristics. In prior research, as pointed out in the above paper, when the roughness of the road surface on which the left and right tires’ travel paths are different, the estimation accuracy of dynamic tire force and bridge frequency is reduced. However, no onboard sensors have been proposed to verify this, and no measurements have been conducted on heavy vehicles capable of exciting bridges. Therefore, this study will develop a new in-vehicle measurement sensor system to measure 3D behavior.

This study focuses on the automatic determination of driving sections on bridges included in vehicle vibrations for the social implementation of drive-by monitoring. A sensor system capable of measuring the three-dimensional behavior of vehicles was proposed and field-tested. By comparing three types of information, namely vehicle vibration, location information measured by GPS sensors installed on the bridge, and location information obtained from Google Maps, the correct label for the segment traveling on the bridge can be obtained. A machine learning method is used to determine whether the vehicle is traveling on the bridge section or not. The discrimination model is a supervised binary classification problem using C-LSTM networks.

## 2. Methodology

### 2.1. Labels for Driving on Bridges

There are three possible ways to label “on a bridge” to vehicle data. The first method uses the data measured by the GPS receiver installed on the vehicle and uses the latitude and longitude information of Google Maps to determine the entry and exit of the bridge. This study adopts Hubeny’s formula to calculate the distance between the bridge entrance/exit line and the vehicle position measured by the GPS receiver at the front wheel. The timings of entry/exit are determined when each distance is minimized.

The second method uses the data measured by the GPS receivers installed on the vehicle and bridge. This method can be more accurate because similar atmospheric conditions have affected the vehicle and bridge GPS receivers simultaneously.

The third method uses the vibration peaks obtained when a vehicle runs on a joint before and after the bridge. When applying this to vehicle response analysis, “on a bridge” data are often extracted concerning the peaks caused by the joints. However, peaks are not always indicating the bridge joints. This study compares these three methods and selects the best way to label the data.

### 2.2. C-LSTM Networks

C-LSTM networks is a deep learning model that combines CNN and RNN. The CNN layer handles spatial relationships and short-time dependencies, reducing the data input to the LSTM layer [19]. Yang et al. [23] applied hierarchical CNN and GRU (HCG), a combination of CNN and GRU (Gated Recurrent Unit), to bridge vibration measurements to detect structural damage directly. According to them, the magnitude of the forces received varies from location to location due to the bridge’s structure, but adjacent locations are subjected to similar forces. Therefore, data measured by adjacent sensors often have similar patterns and are dependent on each other. On the other hand, a signal from one sensor measured simultaneously influences the next signal. Thus, the data are influenced simultaneously by spatial and temporal data, so it is necessary to design an appropriate model to learn and extract spatio-temporal features simultaneously. Since vehicle vibration data are measured at moving points and include vibration of the interaction with bridges, C-LSTM networks are considered effective in capturing the spatio-temporal relationship. However, LSTM is computationally more expensive than GRU because of its greater expressive power. This study aims to address the challenge of identifying the traveled sections on bridges during vehicle vibrations, and the investigation of a more accurate method is a subject for future work.

In this study, the C-LSTM model examined by Kim and Cho (2018) [21] and Kim and Cho (2019) [22] is used as a reference, and it is composed of convolution, activation, pooling, LSTM, and dense layer. Activation is a function that determines the output value for a weighted sum of inputs. Mainly nonlinear functions are used, and ReLu, tanh, and softmax are considered. Pooling is used to extract data features, and MaxPooling is used in this paper. The kernel size, stride size, and activation function were adjusted to minimize the loss function value calculated in the training data. Table 1 shows the finally obtained model. Prediction results and training labels are compared to the learned model, and the percentage of agreement is taken as the percentage of correct responses. Of the data used for analysis, 70% was used for learning, and 30% was used for verification.

## 3. Field Study

This paper deals with the data obtained from real-world environmental tests using a heavy vehicle in Ibaraki Prefecture, Japan. The measurement experiment was conducted from 19–21 July 2021 and 7–9 December 2021.

### 3.1. Experimental Setting

Accelerometers were installed at a total of 10 locations (8 locations (front or rear × left or right × sprung mass or unsprung mass) and two additional locations). A three-axis MEMS (Micro Electro Mechanical Systems) accelerometer (ADXL 355 by Analog Devices Inc., Norwood, MA, USA) was used in the experiment. The sampling frequency is 300 Hz, and the acceleration measurement range is ±8 g. The vehicle position is measured by GPS receivers (AE-GYSFDMAXB by TAIYO TUDEN, Tokyo, Japan) installed at three points (one point directly above the center of the front wheels and two points directly above the left and right rear wheels). The test vehicle (Fighter Mitsubishi FUSO) was loaded with steel plates, weighing 13.6 tons. Figure 1 shows a photograph of the vehicle used in the experiment and the accelerometer’s position. Figure 2 shows the vehicle acceleration sensor system used in this study. The measurement system uses FPGA (Field-Programmable Gate Array), and each accelerometer and GPS receiver are connected using a LAN cable. In addition, power is supplied from the mobile battery, and the data are recorded on the PC. Vehicle vibration and position data are output 300 times per second. The time of vehicle vibration and position is synchronized based on the accurate time (GPS timestamp) included in the satellite signal measured by the GPS receiver. However, the GPS position and time information is updated every second. At the time of the first updated GPS timestamp, the measured CPU time is set to 0. By the above operation, the vibration and position of the vehicle can be accurately synchronized with the GPS time. Since there is a time lag before the measured latitude and longitude are saved, the previously measured latitude and longitude will be output when the GPS timestamp is updated. Therefore, the latitude/longitude data were corrected to match the update timing of the GPS time stamp. Latitude/longitude data with a sampling frequency of 1 Hz were linearly interpolated to be 300 Hz.

### 3.2. GPS Receiver for Bridges and Target Bridges

Figure 3 shows the GPS receiver installed on the bridge. The GPS receiver for bridges consists of a microcomputer (GR-peach by Core Corporation, Tokyo, Japan), a GPS module (ADA-746 by Adafruit, New York, NY, USA), and a GPS antenna. Power is supplied from the mobile battery, and the data are recorded on the SD card. A GPS receiver was installed at each entrance and exit of the bridge (right and left side) to measure the bridge position. The target bridge was a simple support beam in Ibaraki prefecture, Japan, which has a joint. The bridges driven in this study are summarized in Table 2 (bridge names are anonymized). Only the time overlapping with the vehicle measurement data is extracted and linearly interpolated at 300 Hz from the bridge position information data.

Figure 4 shows the vehicle vibration and position data obtained from three runs at the University of Tsukuba. The vehicle vibrations are measured at the unsprung mass at the front-left wheel. The high repeatability is confirmed.

### 3.3. Bridge Description

The target bridges were seven simply supported, single-span bridges in Japan’s Ibaraki Prefecture. Information on the bridges is summarized in Table 2. As a representative example, a photograph of Bridge A is shown in Figure 5. Only the materials are listed for Bridge C because its structure was unknown. There is a joint between the road and the bridge in all other bridges except for Bridge D. The pavement was structurally sound and smooth even though it had been in use for a considerable time.

### 3.4. Preprocessing

In this section, bridge labels are compared. Figure 6 plots the data measured for the position of the vehicle’s front wheels and the unsprung mass of the left side of the front wheels during one run on Bridge A. The black dots indicate the location of bridge corners obtained from Google Maps. As shown in Figure 5a, the bridge corner has a joint, which generates a peak in vehicle vibration when driving. The position of the peak of the vehicle vibration and the position of the bridge corner are misaligned, probably due to an error in the position information measured by the GPS installed in the vehicle. Particularly in urban areas, satellite signals are difficult to receive due to reflections, diffraction, and blockage by nearby obstacles. Figure 6 also visualizes the data measured at the front and rear wheels. From top to bottom: left side unsprung mass, right side unsprung mass, left side unsprung mass, right side unsprung mass, bridge label from Google Maps, bridge label from vehicle vibration peak, and bridge label from bridge GPS. As shown in Figure 7, labeling by vehicle vibration peaks is not a problem if the pavement is smooth. Therefore, labeling is performed in this study by the peak of vehicle vibration when driving on the bridge. However, if potholes or maintenance holes before or after the bridge, vehicle vibration peaks will occur even if they are not at the joints. All data were visually verified to be acceptable to prevent mislabeling of bridges. The analysis data were extracted before and after the bridge so that the data were three times the size of the section on the bridge. Data measured at eight locations (front or rear x left or right x sprung mass or unsprung mass) were used for the measured vehicle vibrations. Vehicle vibration is noise processed before being learned by C-LSTM. Since vehicle vibration includes environmental noise, extracting only the interest data is crucial. Therefore, in this study, a low-pass filter to reduce frequencies above 30 Hz, a high-pass filter to reduce frequencies below 1 Hz, and a band-pass filter to reduce frequencies outside of the 1–30 Hz range were used to compare the results with those without noise processing.

## 4. Result and Discussion

For the data measured in the field test, the detection of the traveling section on the bridge by C-LSTM networks showed high accuracy in both the front and rear models. The results obtained are shown in Table 3. In particular, high accuracy was obtained for the front wheel models when no filter was applied, and the highest accuracy was obtained for the rear wheel models when a low-pass filter was applied. The vehicle used in this experiment was designed to transport crushed stones and other materials, and the tires and suspension of the front wheels were designed to enhance riding comfort. Therefore, the data measured at the front wheels are expected to have reduced low-frequency and high-frequency noise effects. Therefore, the filtering performed as preprocessing had little effect. On the other hand, vibration contamination by high-frequency components was observed, especially in the rear wheels, and it is considered that the low-pass filter made the steep peaks more pronounced when running at joints.

Only bridges with joints were included in this study and were given the label “on a bridge” based on the peaks in the vehicle vibration. However, since some bridges do not have joints, this technique has only limited applicability and needs to be improved to acquire practicality. For example, non-jointed bridges are integral bridge abutments (IABs). While IABs are easy to run and construct, soil settlement occurs as the bridge expands and contracts due to temperature. To mitigate this, Zadehmohamad et al., 2021 [29], considered a backfill mixture of tire material and soil, which is being developed with high interest. To automatically detect non-jointed bridges, it is necessary to study a method to extract only the bridge vibration component from the vehicle vibration and utilize it for bridge labels. However, non-jointed bridges have only short spans and are often destroyed by scour. Therefore, it can be assumed that many non-joint bridges will be monitored based on water level data rather than vehicle vibration data, and different approaches, such as vehicle–bridge simultaneous measurement, can be expected.

In addition, the model considered in this study may respond to peaks in vehicle vibration when traveling outside of joints. Therefore, it is necessary to investigate a method to extract only the bridge vibration component from the vehicle vibration and utilize it for bridge labels. Methods to extract only the bridge vibration component have been proposed by Wang et al. (2018) [15] and Murai et al. (2019) [16]. However, it is necessary to synchronize the positions of the front and rear wheels to remove the road surface irregularity component in the vehicle vibration. This method is challenging to apply when there are measurement errors in position information or high vehicle speed because even minor errors in position synchronization can affect the method. Therefore, assuming that the measured vehicle vibration includes bridge vibration, it may be possible to solve this problem by increasing the complexity of the discriminant model. It is expected to develop a multi-class classification model that considers multiple labeling, such as when driving on bridges, when driving on non-bridges, and when driving on seams. Unsupervised learning methodologies would also be effective. In advance, a model is trained to estimate vehicle vibration during non-bridge driving. When the model is applied to the measured vehicle vibration, if the model does not fit well, the vehicle may be traveling on a bridge.

## 5. Conclusions

In this study, an in-vehicle sensor system that can measure the three-dimensional response was developed, and vehicle vibration data were measured while driving on a bridge in Ibaraki Prefecture, Japan. This is the first paper to address the identification of the traveled sections on bridges in vehicle vibration. The accuracy of bridge labeling of segments traveling on bridges was compared based on vehicle vibration data, location information obtained from Google Maps, and location information measured by a GPS receiver grounded on the bridge. In this study, the peaks generated at the joints before and after the bridge in the vehicle vibration were used; supervised learning with C-LSTM networks was used to perform binary classification, and it was possible to estimate the driving section on the bridge with high accuracy for each model of the front and rear wheels. However, the trained model has a problem in that it responds to vehicle vibration peaks other than joints. Future work includes extracting bridge vibration components from vehicle vibration and labeling them accordingly. This is possible if the axle positions of the front and rear wheels can be synchronized, and the effects of road surface irregularities can be eliminated, but this is challenging because of the effects of GPS receiver measurement errors and vehicle speed. Therefore, it is considered realistic to extend the model to multi-class classification and build a model with multiple labels, such as when driving on and off bridges, when entering bridges, and when exiting bridges. Unsupervised learning is also expected to be used to build models that do not fit well only on sections on the bridge.

Additionally, this study did not attempt to compare the accuracy of the prediction models. Since various C-LSTM models and related models have been proposed in previous studies, comparing them is also a future task. The mechanical relationship of the method to the joints and bridge structures has not been clarified. Numerical simulation is useful to verify this. The proposed model was constructed for the front and rear wheel sections. However, heavy vehicles often have more than just front and rear wheels. Therefore, it is necessary to examine whether a model with the same accuracy can be constructed using a vehicle model other than the one considered in this paper.

## Figures and Tables

**Figure 1 sensors-22-03486-f001:**
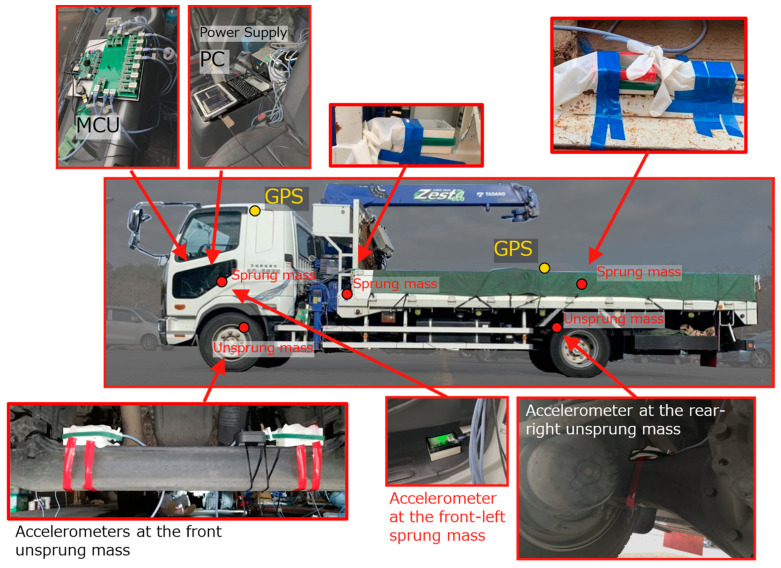
Vehicle and Accelerometers installation point.

**Figure 2 sensors-22-03486-f002:**
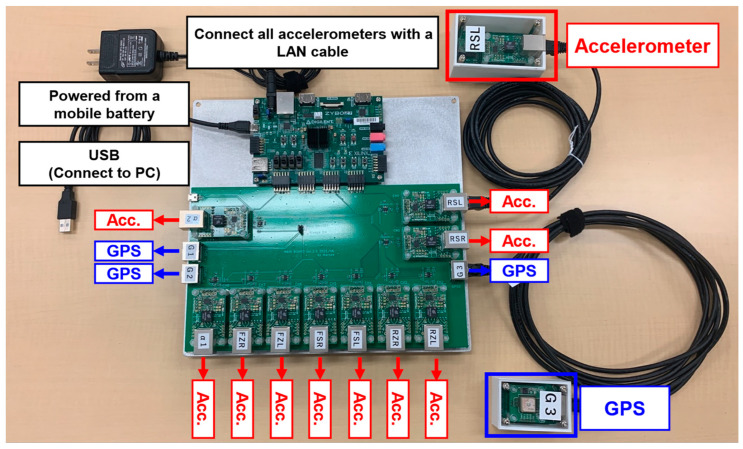
Vehicle measurement sensor system used for measurement.

**Figure 3 sensors-22-03486-f003:**
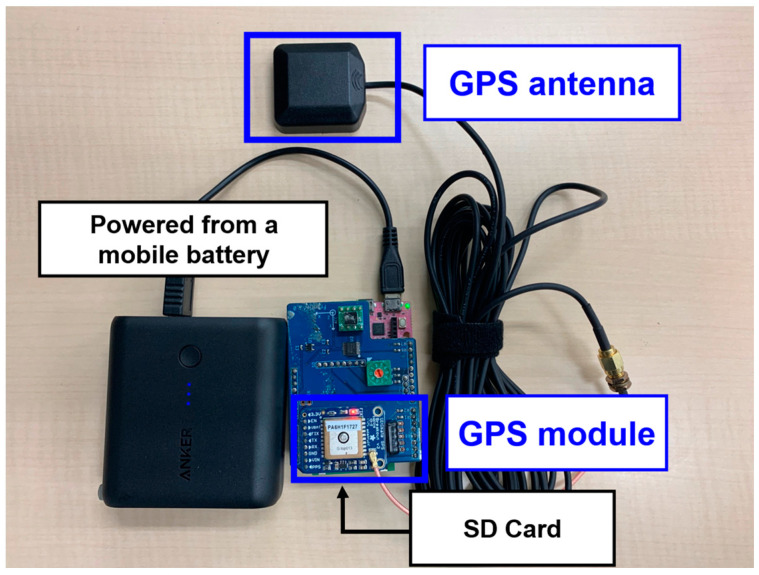
The GPS receiver installed on the bridge.

**Figure 4 sensors-22-03486-f004:**
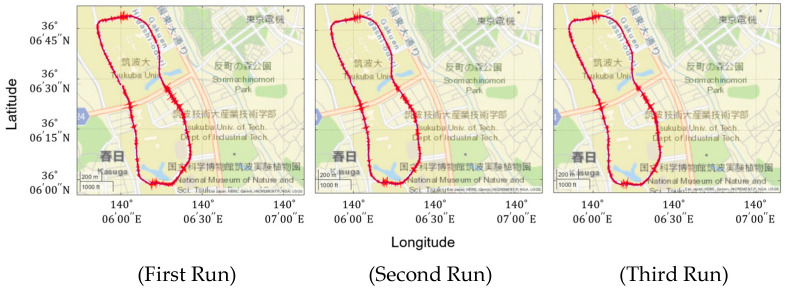
The measured vehicle vibration and position data.

**Figure 5 sensors-22-03486-f005:**
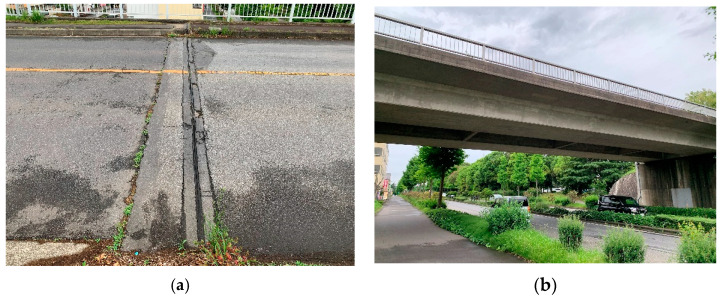
Photo of Bridge A: (**a**) rubber joints for bridges; (**b**) general view.

**Figure 6 sensors-22-03486-f006:**
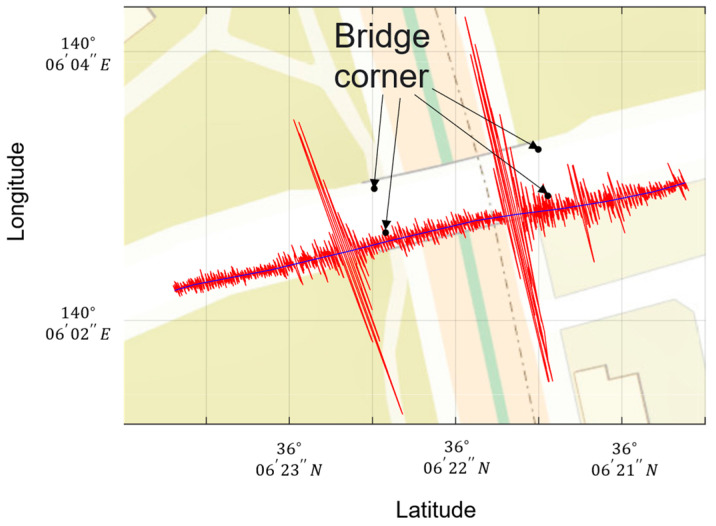
Data measured on the left side of the front wheels for unsprung weight when running Bridge A.

**Figure 7 sensors-22-03486-f007:**
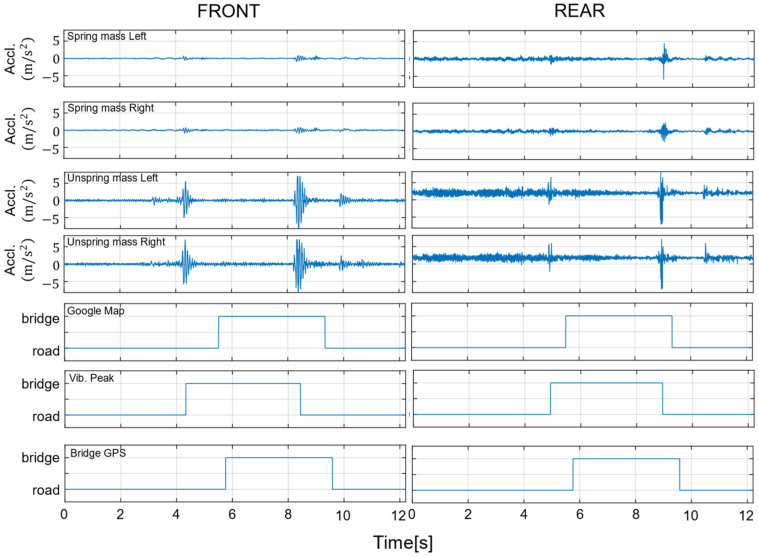
Data measured when driving on Bridge A and bridge labels.

**Table 1 sensors-22-03486-t001:** Architecture of proposed C-LSTM networks.

Layers	Filter	Kernel Size	Stride
Convolution	96	32	1
Activation (ReLu)	-	-	-
MaxPooling	-	4	1
Convolution	96	32	1
Activation (ReLu)	-	-	-
MaxPooling	-	4	1
LSTM (200)	-	-	-
Activation (tanh)	-	-	-
Dense (2)	-	-	-
softmax	-	-	-

**Table 2 sensors-22-03486-t002:** Specifications of the target bridge and number of runs.

Bridge Name	Structure	Joint Type	Bridge Span (m)	Bridge Width (m)	Number of Runs
A	PC Concrete Box Girder	Rubber	30.9	13.0	12
B	PC Concrete T-Girder	Steel	14.0	10.7	6
C	Concrete	Rubber	12.0	11.0	4
D	RC Concrete I-Girder	None	12.6	6.8	6
E	RC Concrete Girder	Rubber	14.0	6.6	5
F	PC Concrete Girder	Rubber	36.8	18.8	1
G	Steel Girder	Rubber	16.0	16.8	4

**Table 3 sensors-22-03486-t003:** Accuracy rate of the proposed model.

	Front Model		Rear Model	
	Train	Test	Train	Test
None	1.000	0.981	0.958	0.835
High-pass Filter	1.000	0.775	1.000	0.820
Low-pass Filter	1.000	0.970	1.000	0.855
Band-pass Filter	1.000	0.864	1.000	0.735

## Data Availability

Not applicable.

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
