# Peer review of "Application of C-LSTM Networks to Automatic Labeling of Vehicle Dynamic Response Data for Bridges"

_sensors, 2022, doi:10.3390/s22093486_

Round 1
Reviewer 1 Report
Reviewer’s comments on “Application of C-LSTM Networks to Automatic Labelling of Vehicle Dynamic Response Data for Bridges” (Ref# sensors-1619874)
Summary:
The authors proposed the useful idea how to diagnosis the structural health of the bridge by monitoring the vehicle vibration data of a moving heavy vehicle. They claim in this paper that the detection of the traveling section on the bridge by C-LSTM Networks showed high accuracy in both the front and rear models. However, how to improve the accuracy of other parts should be considered not just the front and rear parts because heavy vehicle often don't have only front and rear wheels. The authors also claim that the low-pass filter can effectively improve the accuracy but does not explain in detail.
Taking all this into account above, I recommend this review article can be published in the Sensors journal of MDPI in its current form after response my concern. It is recommended that the authors can use Word Editing to assist check the manuscript wording for readers to easy to read as well.
Author Response
Thank you very much for taking time out of your busy schedule to review our paper and offer your advice. We have revised the paper's content about your meaningful comments. We would appreciate it if you could review the paper again.The attached document highlights the changes in red.
Point 1: They claim in this paper that the detection of the traveling section on the bridge by C-LSTM Networks showed high accuracy in both the front and rear models. However, how to improve the accuracy of other parts should be considered not just the front and rear parts because heavy vehicle often don't have only front and rear wheels.
Thank you for your comment. Numerical simulations and new experiments are needed to verify the points you have pointed out. For this reason, we have added the following to this paper as a future issue.
LNSA (Line number of the scientific article) 332-334
“However, heavy vehicles often have more than just front and rear wheels. Therefore, it is necessary to examine whether a model with the same accuracy can be constructed using a vehicle model other than the one considered in this paper.”
Point 2: The authors also claim that the low-pass filter can effectively improve the accuracy but does not explain in detail.
Based on your suggestion, we have revised the explanation for each of the front and rear wheel models.
LNSA 282-291
“In particular, high accuracy was obtained for the front wheel models when no filter was applied, and the highest accuracy was obtained for the rear wheel models when a low-pass filter was applied. The vehicle used in this experiment was designed to transport crushed stones and other materials, and the tires and suspension of the front wheels were designed to enhance riding comfort. Therefore, the data measured at the front wheels are expected to have reduced low-frequency and high-frequency noise effects. Therefore, the filtering performed as preprocessing had little effect. On the other hand, vibration contamination by high-frequency components was observed, especially in the rear wheels, and it is considered that the low-pass filter made the steep peaks more pronounced when running at joints.”
Point 3: It is recommended that the authors can use Word Editing to assist check the manuscript wording for readers to easy to read as well.
Using the Grammarly app, I corrected the English in the paper.

Reviewer 2 Report
The authors presented a study on application of C-LSTM networks o automatic labelling of vehicle dynamic response data for bridges. The topic is highly interesting, however the manuscript was written in brief. Much more contents need to be included before its publication.
(1) The C-LSTM network is a key point the authors want to make, however the section on C-LSTM only has a few sentences. The training and validation results need to be included and discussed.
(2) Three methods are mentioned on page 3 of the manuscript. A comparison of the three methods on the accuracy of "on the bridge" labeling will be interesting to see.
(3) Low pass and High pass filters are used, but are not given in the manuscript. Please include details of these filters and show their respective results, and then discuss the accuracy of each filter on the labeling accuracy of "on the bridge".
(4) There are some grammar mistakes and some sentence does not read smooth and make sense, such as line 116-118. What does it mean by "as a section traveling over a bridge"?
Author Response
Thank you very much for taking time out of your busy schedule to review our paper and offer your advice. We have revised the paper's content about your meaningful comments. We would appreciate it if you could review the paper again. The attached document highlights the changes in red.
Point 1: The C-LSTM network is a key point the authors want to make, however the section on C-LSTM only has a few sentences. The training and validation results need to be included and discussed.
Thank you for your comment. As you pointed out, I did not explain myself well enough. I have added the following.
LNSA (Line number of the scientific article) 175-191
“C-LSTM Networks is a deep learning model that combines CNN and RNN. The CNN layer handles spatial relationships and short-time dependencies, reducing the data input to the LSTM layer[18]. Yang et al. [22] applied Hierarchical CNN and GRU (HCG), a combination of CNN and GRU(Gated Recurrent Unit), to bridge vibration measurements to detect structural damage directly. According to them, the magnitude of the forces received varies from location to location due to the bridge's structure, but adjacent locations are subjected to similar forces. Therefore, data measured by adjacent sensors often have similar patterns and are dependent on each other. On the other hand, a signal from one sensor measured simultaneously influences the next signal. Thus, the data are influenced simultaneously by spatial and temporal data, so it is necessary to design an appropriate model to learn and extract Spatio-Temporal features simultaneously. Since vehicle vibration data is measured at moving points and includes vibration of the interaction with bridges, C-LSTM Networks are considered effective in capturing the Spatio-Temporal relationship. However, LSTM is computationally more expensive than GRU because of its greater expressive power. This study aims to address the challenge of identifying the traveled sections on bridges during vehicle vibrations, and the investigation of a more accurate method is a subject for future work.”
Point 2: Three methods are mentioned on page 3 of the manuscript. A comparison of the three methods on the accuracy of "on the bridge" labeling will be interesting to see.
Thank you for pointing this out. Front and rear wheel suspensions are different and may affect the accuracy of "on the bridge" labeling. However, this study is intended to be a starting point for the new problem of identifying the section of a vehicle traveling on a bridge during vehicle vibration. Therefore, we will not discuss this point in this study, but would like to address it in the future.
Point 3: Low pass and High pass filters are used, but are not given in the manuscript. Please include details of these filters and show their respective results, and then discuss the accuracy of each filter on the labeling accuracy of "on the bridge".
We have added the following for better clarity. As mentioned in point 2, the percentage of correct answers for the label "on the bridge" is an issue to be addressed in the future.
LNSA (Line number of the scientific article) 273-277
“Vehicle vibration is noise processed before being learned by C-LSTM. Since vehicle vibration includes environmental noise, extracting only the interest data is crucial. Therefore, in this study, a LowPass Filter to reduce frequencies above 30 Hz, a HighPass Filter to reduce frequencies below 1 Hz, and a BandPass Filter to reduce frequencies outside of the 1-30 Hz range were used to compare the results with those without noise processing.”
Point 4: There are some grammar mistakes and some sentence does not read smooth and make sense, such as line 116-118. What does it mean by "as a section traveling over a bridge"?
Thank you for your suggestion. To make the paper more readable, we have modified it by referring to an application named Grammarly.

Reviewer 3 Report
Reviewed post Application of C-LSTM Networks to Automatic Labeling of Vehicle Dynamic Response Data for Bridges, rated as good. However, the article seems to me more like a quality article at a conference than in a prestigious scientific journal, but I rate its content as a contribution with great potential to reach a wider range of readers, including experts in the field of pavement engineering. For the purposes of rating an article as very good or excellent, I would like to make the following requirements and recommendations.
Mandatory requirements:
LNSA (Line number of the scientific article) 8-14 ... Abstract: The abstract needs to be reworked to more specifically describe the issue from a scientific point of view. He is written too generally (The maintenance and management of bridges that support road infrastructures are essential because it affects the economy and human lives ... measured by a heavy vehicle ... with high accuracy ...) and therefore he needs to "breathe" higher technical dimension.
LNSA 11… model using C-LSTM Networks.. I recommend explaining the abbreviation when it is first used, especially because many readers of the abstract working in other scientific fields than the authors do not know the abbreviation in question.
LNSA 118-118 ... I consider the scope of Introduction (100 lines) to be unbalanced in scope to the total length of the article (258 lines). I recommend shortening it and expanding the technical data related to the topic of the paper, which I write about in other parts of the review. In this part of the article, 4 works by Yamamoto K. are cited, which is not standard, especially to the total number of references 25. I recommend selecting the 2 most important works by Yamamoto K and adding other relevant scientific information sources.
LNSA 119-143… Methodology (24 lines) in contrast to Introduction is too brief. It is necessary to add a separate subchapter devoted to a brief description of the bridge structures of interest. Photographs of a typical bridge object (s) would also be appropriate, including detailed views of the joint as a relevant source of induced vehicle dynamic response peaks.
LNSA 189… Figure 4. The measured vehicle vibration and position data ... I recommend reworking this image as one of the focal points of the paper (eg dividing it into 3 separate images, ...). It is necessary to add at least an indicative scale of vibration acceleration to the marked vibration waveforms and to supplement the data on relevant road sources inducing vibration extremes.
LNSA 228… Table 3. Accuracy rate of the proposed model ... The format of this table needs to be reworked and aligned with the other tables of the article and with the requirements of the journal.
LNSA 273… pan (online), September 14-17, 2021, ... the comma at the end of the sentence must be replaced by a dot.
LNSA 288… Hochreiter, S .; Schmidhuber, J .; Long short-term memory. Neural computation 1997, 9, 8, 1735-1780 ... A dot is needed at the end of the above Reference.
Facultative recommendations:
I encourage authors to consider changing the title of the article to, for example, Analysis of Pavement Performance considering Dynamic Axle Load Spectra induced longitudinal unevenness.
LNSA 186…Table 2. Specifications of the target bridge and number of runs ... The table format is not compatible with the standard table formats in scientific journals and I also recommend adding physical units and considering adding basic information about type used joints as a source of crucial oscillations.
LNSA 208… Figure 5. Data obtained when driving on A bridge and bridge label...or would it be better to use sprung and unsprung mass in the right part of the picture as opposed to spring and unspring?
LNSA 228… Table 1 Architecture of proposed C-LSTM Networks ... It would be appropriate to reformat the table and add a brief explanation of the items Activation: ReLu, tanh…
LNSA 229-242… 5. Conclusions… The conclusions need to be reworked and expanded, supplemented with relevant information not only on the method of measurement but also on bridge structures and unevenness pavement level as a decisive source of vibration. In the event that there are equality measurements for the above-mentioned road sections by means of the world-established parameter IRI (International Roughness Index), it is appropriate to state specific values.
In conclusion, I rate the article as good, but based on my scientific experience, I state that it needs to be improved so that it can be presented in a renowned scientific journal. The paper presents several interesting findings, based on exact experiments, but I miss their more detailed application in the field of road engineering. I reviewed article as a contribution with significant potential for improvement, and therefore, if the presented recommendations are incorporated, I am able to process a re-review within 3 days.
Author Response
Thank you very much for taking time out of your busy schedule to review our paper and offer your advice. We have revised the paper's content about your meaningful comments. We would appreciate it if you could review the paper again. The attached document highlights the changes in red.
Point 1:
LNSA (Line number of the scientific article) 8-14 ... Abstract: The abstract needs to be reworked to more specifically describe the issue from a scientific point of view. He is written too generally (The maintenance and management of bridges that support road infrastructures are essential because it affects the economy and human lives ... measured by a heavy vehicle ... with high accuracy ...) and therefore he needs to "breathe" higher technical dimension.
Thank you for your comment. As you indicated, we have made the following corrections.
LNSA 14-27
“Maintaining bridges that support road infrastructure is critical to the economy and human life. Structural health monitoring of bridges using vibration includes direct monitoring and drive-by monitoring. Drive-by monitoring uses a vehicle equipped with accelerometers to drive over bridges and estimates the bridge's health from the vehicle vibration obtained. In this study, we attempt to identify the driving segments on bridges in the vehicle vibration data for the practical application of Drive-by Monitoring. We have developed an in-vehicle sensor system that can measure three-dimensional behavior, and we propose a new problem of identifying the driving segment of vehicle vibration on a bridge from data measured in a Field experiment. A supervised binary classification model using C-LSTM(Convolution - Long Term Short Memory) Networks was constructed and applied to data measured, and the model was successfully constructed with high accuracy. The model developed in this study may respond to the peaks due to joints before and after the bridge. Therefore, future work is needed to propose a running label on bridges based on bridge vibration and to extend the model to a multi-discriminant model.”
Point 2:
LNSA 11… model using C-LSTM Networks.. I recommend explaining the abbreviation when it is first used, especially because many readers of the abstract working in other scientific fields than the authors do not know the abbreviation in question.
Thank you for your comment. we have corrected the missing points in addition to C-LSTM.
Point 3:
LNSA 118-118 ... I consider the scope of Introduction (100 lines) to be unbalanced in scope to the total length of the article (258 lines). I recommend shortening it and expanding the technical data related to the topic of the paper, which I write about in other parts of the review. In this part of the article, 4 works by Yamamoto K. are cited, which is not standard, especially to the total number of references 25. I recommend selecting the 2 most important works by Yamamoto K and adding other relevant scientific information sources.
Other reviewers pointed out the lack of an introduction, and we have also addressed this issue. The introduction is 157 lines out of a total of 334 lines. We have reduced the number of papers by K. Yamamoto to two as you suggested.
We have added the following for better clarity. As mentioned in point 2, the percentage of correct answers for the label "on the bridge" is an issue to be addressed in the future.
LNSA (Line number of the scientific article) 274-278
“Vehicle vibration is noise processed before being learned by C-LSTM. Since vehicle vibration includes environmental noise, extracting only the interest data is crucial. Therefore, in this study, a LowPass Filter to reduce frequencies above 30 Hz, a HighPass Filter to reduce frequencies below 1 Hz, and a BandPass Filter to reduce frequencies outside of the 1-30 Hz range were used to compare the results with those without noise processing.”
Point 4:
LNSA 119-143… Methodology (24 lines) in contrast to Introduction is too brief. It is necessary to add a separate subchapter devoted to a brief description of the bridge structures of interest. Photographs of a typical bridge object (s) would also be appropriate, including detailed views of the joint as a relevant source of induced vehicle dynamic response peaks.
Thank you very much for your very helpful remarks. 3.3 Bridge Description" and Figure 5(page. 8) are the relevant sections. I have added the following note.
LNSA 246-252
“3.3. Bridge Description The target bridges were seven simply supported, single-span bridges in Japan Ibaraki Prefecture. Information on the bridges is summarized in Table 2. As a repre-sentative example, a photograph of A Bridge is shown in Figure 5. Only the materials are listed for the bridge C because its structure was unknown. There is a joint between the road and the bridge in all other bridges, except for Bridge D. The pavement was structurally sound and smooth, even though it had been in use for a considerable time.”
Point 5:
LNSA 189… Figure 4. The measured vehicle vibration and position data ... I recommend reworking this image as one of the focal points of the paper (eg dividing it into 3 separate images, ...). It is necessary to add at least an indicative scale of vibration acceleration to the marked vibration waveforms and to supplement the data on relevant road sources inducing vibration extremes.
Thank you for your comment. Since the vehicle vibration and bridge labels are time-synchronized, we split them into two separate images. The waveforms in Figure 6(page. 9) are plotted perpendicular to the direction of travel of the vehicle's front wheels. The waveforms have been adjusted in amplitude, easy to read. For this reason, we chose not to introduce a scale in this case.
LNSA 254-266
“Figure 6 plots the data measured for the position of the vehicle's front wheels and the unsprung mass of the left side of the front wheels during one run on Bridge A. The black dots indicate the location of bridge corners obtained from Google Maps. As shown in Figure 5(a), the bridge corner has a joint, which generates a peak in vehicle vibration when driving. The position of the peak of the vehicle vibration and the position of the bridge corner are misaligned, probably due to an error in the position information measured by the GPS installed in the vehicle. Particularly in urban areas, satellite signals are difficult to receive due to reflections, diffraction, and blockage by nearby obstacles. Figure 6 also visualizes the data measured at the front and rear wheels. From top to bottom: left side unsprung mass, right side unsprung mass, left side unsprung mass, right side unsprung mass, bridge label from GoogleMap, bridge label from vehicle vibration peak, and bridge label from bridge GPS. As shown in Figure 7, labeling by vehicle vibration peaks is not a problem if the pavement is smooth.”
Point 6:
LNSA 228… Table 3. Accuracy rate of the proposed model ... The format of this table needs to be reworked and aligned with the other tables of the article and with the requirements of the journal.
We have revised the table based on your suggestion and the tables in other papers(page. 10).
Point 7:
LNSA 273… pan (online), September 14-17, 2021, ... the comma at the end of the sentence must be replaced by a dot.
The correction has been made as you indicated.
Point 8:
LNSA 288… Hochreiter, S .; Schmidhuber, J .; Long short-term memory. Neural computation 1997, 9, 8, 1735-1780 ... A dot is needed at the end of the above Reference.
The correction has been made as you indicated.
Point 9:
I encourage authors to consider changing the title of the article to, for example, Analysis of Pavement Performance considering Dynamic Axle Load Spectra induced longitudinal unevenness.
As we discussed, the subject of this study is to identify the traveled sections on the bridge that are included in the vehicle vibration. Therefore, we do not intend to change our current title, although your remarks were beneficial.
Point 10:
LNSA 186…Table 2. Specifications of the target bridge and number of runs ... The table format is not compatible with the standard table formats in scientific journals and I also recommend adding physical units and considering adding basic information about type used joints as a source of crucial oscillations.
Thank you for your comment. I have modified the table as you suggested, and I think it is much clearer now(page. 11).
Point 11:
LNSA 208… Figure 5. Data obtained when driving on A bridge and bridge label...or would it be better to use sprung and unsprung mass in the right part of the picture as opposed to spring and unspring?
The correction has been made as you indicated.
Point 12:
LNSA 228… Table 1 Architecture of proposed C-LSTM Networks ... It would be appropriate to reformat the table and add a brief explanation of the items Activation: ReLu, tanh…
Thank you for your comment. In addition to the corrections to the table you suggested, I have added explanations regarding Activation and Pooling.
LNSA 194-196
“Activation is a function that determines the output value for a weighted sum of inputs. Mainly nonlinear functions are used, and ReLu, tanh, and softmax are considered. Pooling is used to extract data features, and MaxPooling is used in this paper.”
Point 13:
LNSA 229-242… 5. Conclusions… The conclusions need to be reworked and expanded, supplemented with relevant information not only on the method of measurement but also on bridge structures and unevenness pavement level as a decisive source of vibration. In the event that there are equality measurements for the above-mentioned road sections by means of the world-established parameter IRI (International Roughness Index), it is appropriate to state specific values.
This study assumes that the model will apply to all simply supported bridges, regardless of the bridge structure, joint type, etc. However, as you say, the non-uniformity of pavement levels, etc., which is the decisive cause of vibration, needs to be considered in the future. Therefore, We have added the following as future tasks for these investigations.
LNSA 329-331
“The mechanical relationship of the method to the joints and bridge structures has not been clarified. Numerical simulation is useful to verify this.”

Reviewer 4 Report
Please further elaborate on the novelty of your work in abstract.
Please further describe the main steps that you followed and the main outstanding outcomes in the abstract.
The presented introduction is pretty modest. Please include a brief but critical review regarding the conducted research studies in the introduction. It is recommended to add a section “research significance” and highlight the main contribution of your findings.
Please include the latest research studies related to your work preferably between 2019 and 2022
Please include a brief summary of using recycled material on improving the long-term response of bridge structures. Accordingly, please mention the advantages of using the article titled “Physical modelling of the long-term behavior of integral abutment bridge backfill reinforced with tire-rubber” in your research.
Please elaborate on the advantages and short-comings of the C-LSTM network
The followed methodology should be revised. Please perform numerical simulation to verify the proposed results followed by the developed model by Kim and Cho.
Please elaborate on the limitation of the processing process (chapter 3.3)
Please revise the discussion and include a comparative analysis on stated outcomes and the available relevant studies. It is recommended to include a statistical analysis on the main outcomes.
Please quantify your findings and present them in conclusion. The current version of the conclusion is simply descriptive.
Author Response
Thank you very much for taking time out of your busy schedule to review our paper and offer your advice. We have revised the paper's content about your meaningful comments. We would appreciate it if you could review the paper again. The attached document highlights the changes in red.
Point 1:
Please further elaborate on the novelty of your work in abstract. Please further describe the main steps that you followed and the main outstanding outcomes in the abstract.
Thank you for your comment. We have revised the abstract based on your suggestion, and it is now easier to understand.
LNSA (Line number of the scientific article) 14-27
“Maintaining bridges that support road infrastructure is critical to the economy and human life. Structural health monitoring of bridges using vibration includes direct monitoring and drive-by monitoring. Drive-by monitoring uses a vehicle equipped with accelerometers to drive over bridges and estimates the bridge's health from the vehicle vibration obtained. In this study, we attempt to identify the driving segments on bridges in the vehicle vibration data for the practical application of Drive-by Monitoring. We have developed an in-vehicle sensor system that can measure three-dimensional behavior, and we propose a new problem of identifying the driving segment of vehicle vibration on a bridge from data measured in a Field experiment. A supervised binary classification model using C-LSTM(Convolution - Long Term Short Memory) Networks was constructed and applied to data measured, and the model was successfully constructed with high accuracy. The model developed in this study may respond to the peaks due to joints before and after the bridge. Therefore, future work is needed to propose a running label on bridges based on bridge vibration and to extend the model to a multi-discriminant model.”
Point 2:
The presented introduction is pretty modest. Please include a brief but critical review regarding the conducted research studies in the introduction. It is recommended to add a section “research significance” and highlight the main contribution of your findings. Please include the latest research studies related to your work preferably between 2019 and 2022
Thank you for your comment. Based on your suggestion, we have added the following sentence.
LNSA 129-157
“Wang et al. (2019) [24] proposed estimating tire ground forces with relatively high accuracy even at vehicle speeds of about 30 km/h using a Kalman filter based on vehicle vibrations measured with an iPod installed on the vehicle. Yang et al. (2020)[25] compared the accuracy of estimating the natural frequencies of bridges using data measured by sensors installed on special vehicles and directly measured bridge vibration. When traveling at relatively low speeds, it is possible to estimate up to the second-order natural frequencies of bridges without being affected by the frequency of the vehicle by determining the ground point force from the vehicle vibration. Yang et al. (2022)[26] propose a comprehensive indirect method for estimating girder bridges' fundamental mode dynamic properties with two identical trailer-mounted passing tractors and evaluating the element stiffness. They have been validated both numerically and in field experiments. Locke et al. (2022)[27] compared the system identification capabilities of different OMA(Operational Modal Analysis) techniques on bridges less than 18.28 m (60 ft). The experimental results demonstrate that OMA techniques can be utilized to correctly identify the frequencies of short-span bridges in the dynamic response of a passenger vehicle, even in the presence of time-varying and nonlinear system characteristics.In prior research, as pointed out in the above paper, when the roughness of the road surface on which the left and right tires travel differ, the estimation accuracy of dynamic tire force and bridge frequency is reduced. However, no onboard sensors have been proposed to verify this, and no measurements have been conducted on heavy vehicles capable of exciting bridges. Therefore, this study will develop a new in-vehicle measurement sensor system to measure 3D behavior.
This study proposes a sensor system that can measure the three-dimensional behavior of vehicles and conducts field tests. This paper is the first to identify segments where vehicle vibration includes bridge vibration. The correct label for the section traveling on the bridge is obtained by comparing three types of information: vehicle vibration, location information measured by a GPS sensor installed on the bridge, and location information obtained from Google Maps. The objective is to determine whether the measured acceleration response of a moving vehicle is traveling on a bridge section or not. The discriminant model is a supervised binary classification problem using C-LSTM Networks.”
Point 3:
Please include a brief summary of using recycled material on improving the long-term response of bridge structures. Accordingly, please mention the advantages of using the article titled “Physical modelling of the long-term behavior of integral abutment bridge backfill reinforced with tire-rubber” in your research.
The paper you introduced is one that We are not familiar with, and We learned a lot from it, thank you very much. However, the purpose of this study was to propose a model that can be used for all bridges that are simply supported beams, and the reference to joints and bridge materials is a topic for future research. Therefore, the following text has been added.
LNSA 329-331
“The mechanical relationship of the method to the joints and bridge structures has not been clarified. Numerical simulation is useful to verify this.”
Point 4:
Please elaborate on the advantages and short-comings of the C-LSTM network
Thank you for your comment. We have added the following note.
LNSA 175-191
“C-LSTM Networks is a deep learning model that combines CNN and RNN. The CNN layer handles spatial relationships and short-time dependencies, reducing the data input to the LSTM layer[18]. Yang et al. [22] applied Hierarchical CNN and GRU (HCG), a combination of CNN and GRU(Gated Recurrent Unit), to bridge vibration measurements to detect structural damage directly. According to them, the magnitude of the forces received varies from location to location due to the bridge's structure, but adjacent locations are subjected to similar forces. Therefore, data measured by adjacent sensors often have similar patterns and are dependent on each other. On the other hand, a signal from one sensor measured simultaneously influences the next signal. Thus, the data are influenced simultaneously by spatial and temporal data, so it is necessary to design an appropriate model to learn and extract Spatio-Temporal features simultaneously. Since vehicle vibration data is measured at moving points and includes vibration of the interaction with bridges, C-LSTM Networks are considered effective in capturing the Spatio-Temporal relationship. However, LSTM is computationally more expensive than GRU because of its greater expressive power. This study aims to address the challenge of identifying the traveled sections on bridges during vehicle vibrations, and the investigation of a more accurate method is a subject for future work.”
Point 5:
The followed methodology should be revised. Please perform numerical simulation to verify the proposed results followed by the developed model by Kim and Cho.
Thank you for your comment. As mentioned in Point 3, the mechanical effects of bridge structure, joints, etc. are an issue to be addressed in the future. Also, as you pointed out, I think it is important to compare this model with the deep learning models that have been proposed. However, the purpose of this study is to propose a new problem of identifying the travel section on a bridge during vehicle vibration. Therefore, we decided to compare multiple models and search for a model with better accuracy as a future task.
Point 6:
Please elaborate on the limitation of the processing process (chapter 3.3)
The vehicle's position is based on the location information measured by GPS and therefore contains errors. Therefore, labeling is based on location information measured by Google Maps or GPS receivers installed on bridges. We have added the following note on this matter.
LNSA 254-266
“Figure 6 plots the data measured for the position of the vehicle's front wheels and the unsprung mass of the left side of the front wheels during one run on Bridge A. The black dots indicate the location of bridge corners obtained from Google Maps. As shown in Figure 5(a), the bridge corner has a joint, which generates a peak in vehicle vibration when driving. The position of the peak of the vehicle vibration and the position of the bridge corner are misaligned, probably due to an error in the position information measured by the GPS installed in the vehicle. Particularly in urban areas, satellite signals are difficult to receive due to reflections, diffraction, and blockage by nearby obstacles. Figure 6 also visualizes the data measured at the front and rear wheels. From top to bottom: left side unsprung mass, right side unsprung mass, left side unsprung mass, right side unsprung mass, bridge label from GoogleMap, bridge label from vehicle vibration peak, and bridge label from bridge GPS. As shown in Figure 7, labeling by vehicle vibration peaks is not a problem if the pavement is smooth.”
Point 7:
Please revise the discussion and include a comparative analysis on stated outcomes and the available relevant studies. It is recommended to include a statistical analysis on the main outcomes. Please quantify your findings and present them in conclusion. The current version of the conclusion is simply descriptive.
Thank you for your comment. This study proposes a method for identifying the traveled sections on bridges during vehicle vibration, which has not been addressed in the past. The contribution of this study is limited to suggesting that the objectives of this study can be achieved with data measured in a real environment. Therefore, we consider it difficult to make comparisons with previous studies. As you indicated, we considered that objective evaluation of the obtained results is a matter to be considered in the future.

Round 2
Reviewer 2 Report
The authors have addressed all my comments. The manuscript could be accepted for publications.
Author Response
Thank you for taking time out of your busy schedule to review this paper. As I am not a native English speaker, I apologize for any inconvenience caused by the English expressions. I am interested in reducing the burden of maintenance and improving the safety and reliability of infrastructure structures. I believe that I have made my living thanks to those who have built robust infrastructures. In return, I want to pass on the same environment to the next generation's children. The reviewers' opinions have really improved my paper and helped me to grow as a researcher. Again, thank you very much for your peer review.

Reviewer 3 Report
Based on a detailed study of the second version of the evaluated scientific contribution, in conjunction with a detailed control of the extent of implementation of my recommendations, I would like to state the following facts. The authors have made full use of the potential to improve their contribution, making the version considered a quality scientific contribution to me. For me, as an expert working mainly in the field of pavement mechanics, objectification of the negative environmental impacts induced by road transport, the assessed contribution is an inspiring article. In conclusion, I would like to congratulate the authors on a quality scientific contribution converging with my holistic concept of sustainable design, construction and management of roads implemented at a reasonable price in reasonable quality.
Author Response
Thank you for taking time out of your busy schedule to review this paper. As I am not a native English speaker, I apologize for any inconvenience caused by the English expressions. The paper is now much better than it was before the reviewer's suggestion. Thank you very much for the high evaluation of this research. Thanks to the warm encouragement of the reviewers, I was able to overcome the painful peer review revisions. Through this peer review process, I felt that I grew as a researcher, and above all, I reaffirmed that research is something I enjoy. I am happy that I took up the challenge of this journal. Again, thank you very much for your peer review.

Reviewer 4 Report
Some of the raises concerns are not addressed properly.
The novelty of the work is still not appreciated. The statement on the contribution of this work is generic and cannot address any major concerns in the investigated field.
The suggested article Physical modelling of the long-term behavior of integral abutment bridge backfill reinforced with tire-rubber provides important info on sustainable application of using recycled material to improve the behavior of bridges. The behavior of bridges also concerns dynamic behavior of them and highlighting the importance of the provided method in this article would enhance the quality of the provided introduction by authors.
The main issues (short comings) of the presented work is yet to be addressed. The provided response is insufficient.
Again, the limitations that can hinder the presented work to reach accurate results should be critically discussed.
The authors still have not provided compelling discussion.
Author Response
Thank you very much for providing important comments. We are thankful for the time and energy you expended. Our responses to the referees’ comments are as follow. The attached document highlights the changes in red.
Point 1:
The novelty of the work is still not appreciated. The statement on the contribution of this work is generic and cannot address any major concerns in the investigated field.
Thank you for your comment. The claim of novelty was ambiguous in the text we submitted. Therefore, we have revised it as follows. The abstract has been slightly modified accordingly.
LNSA (Line number of the scientific article) 80-96
Although many papers explicitly state that they measure vehicle vibration, many also assume the use of vehicle position in their signal processing [7,8]. Satellite positioning, navigation, and timing systems, the so-called GPS (Global Positioning System), can measure a vehicle's position. However, the readily available GPS devices are also subject to significant errors, so a vehicle's travel position is not reliable. Bridge locations can be obtained from GIS (Geographic Information System) data, but the resolution is too coarse. Accurate information on the location of each bridge is usually unknown or not digitized. Furthermore, drive-by monitoring primarily uses data only while a vehicle passes over a bridge. Because the time that a vehicle travels on a bridge is so short, the location information measured by GPS devices is limited to a rough extraction of interest data. Therefore, when driving data on bridges is extracted from GPS-only location data, it is likely to include data from sections of the bridge that are not bridges, and this will likely harm the results of the analysis. In addition, multiple runs are a promising approach for drive-by monitoring to overcome the problems of road surface irregularities, environmental effects, and limited measurement time [9]. Therefore, for the social implementation of drive-by monitoring, it is essential to have a technology that can automatically extract only the target data from considerable driving data.
LNSA 156-164
This study focuses on the automatic determination of driving sections on bridges included in vehicle vibrations for the social implementation of drive-by monitoring. A sensor system capable of measuring the three-dimensional behavior of vehicles was proposed and field-tested. By comparing three types of information: vehicle vibration, location information measured by GPS sensors installed on the bridge, and location information obtained from Google Maps, the correct label for the segment traveling on the bridge can be obtained. A machine learning method is used to determine whether the vehicle is traveling on the bridge section or not. The discrimination model is a supervised binary classification problem using C-LSTM Networks.
LNSA 14-27
Maintaining bridges that support road infrastructure is critical to the economy and human life. Structural health monitoring of bridges using vibration includes direct monitoring and drive-by monitoring. Drive-by monitoring uses a vehicle equipped with accelerometers to drive over bridges and estimates the bridge's health from the vehicle vibration obtained. In this study, we attempt to identify the driving segments on bridges in the vehicle vibration data for the practical application of Drive-by Monitoring. We have developed an in-vehicle sensor system that can measure three-dimensional behavior, and we propose a new problem of identifying the driving segment of vehicle vibration on a bridge from data measured in a Field experiment. The "On a bridge" label was assigned based on the peaks in the vehicle vibration when running at joints. A supervised binary classification model using C-LSTM(Convolution - Long Term Short Memory) Networks was constructed and applied to data measured, and the model was successfully constructed with high accuracy. The challenge is to build a model that can be applied to bridges where joints do not exist. Therefore, future work is needed to propose a running label on bridges based on bridge vibration and extend the model to a multi-class model.
Point 2:
The suggested article Physical modelling of the long-term behavior of integral abutment bridge backfill reinforced with tire-rubber provides important info on sustainable application of using recycled material to improve the behavior of bridges. The behavior of bridges also concerns dynamic behavior of them and highlighting the importance of the provided method in this article would enhance the quality of the provided introduction by authors.
Point 3:
The main issues (short comings) of the presented work is yet to be addressed. The provided response is insufficient. Again, the limitations that can hinder the presented work to reach accurate results should be critically discussed. The authors still have not provided compelling discussion.
I have re-read the paper you referred to and am once again aware of the limitations of our technology. Points 2 and 3 have been corrected together. Therefore, we have revised it as follows.
LNSA 296-326
Only bridges with joints existed in the path of interest for this experiment, and were given the label "On-Bridge" based on the peaks in the vehicle vibration. However, since some bridges do not have joints, this technique has only limited applicability and needs to be improved to acquire practicality. For example, non-jointed bridges are integral bridge abutments (IABs). While IABs are easy to run and construct, soil settlement occurs as the bridge expands and contracts due to temperature. To mitigate this, Zadehmohamad et al. 2021 are considering a backfill mixture of tire material and soil, which is being developed with high interest. To automatically detect non-jointed bridges as well, it is necessary to study a method to extract only the bridge vibration component from the vehicle vibration and utilize it for bridge labels. However, non-jointed bridges have only short spans and are often destroyed by scour. Therefore, it can be assumed that many non-joint bridges will be monitored based on water level data rather than vehicle vibration data, and different approaches such as vehicle-bridge simultaneous measurement can be expected.
In addition, the model considered in this study may respond to peaks in-vehicle vibration when traveling outside of joints. Therefore, it is necessary to investigate a method to extract only the bridge vibration component from the vehicle vibration and utilize it for bridge labels. Methods to extract only the bridge vibration component have been proposed by Wang et al. (2018)[14] and Murai et al. (2019)[15]. However, it is necessary to synchronize the positions of the front and rear wheels to remove the road surface irregularity component in the vehicle vibration. This method is challenging to apply when there are measurement errors in position information or high vehicle speed because even minor errors in position synchronization can affect the method. Therefore, assuming that the measured vehicle vibration includes bridge vibration, it may be possible to solve this problem by increasing the complexity of the discriminant model. It is expected to develop a multi-class classification model that considers multiple labeling, such as when driving on bridges, when driving on non-bridges, and when driving on seams. Unsupervised learning methodologies would also be effective. In advance, a model is trained to estimate vehicle vibration during non-bridge driving. When the model is applied to the measured vehicle vibration, if the model does not fit well, the vehicle may be traveling on a bridge.
